# *Habit-DisHabit* Design with a Quadratic Equation: A Better Model of the Hemodynamic Changes in Preschoolers during the Dimension Change Card Sorting Task

**DOI:** 10.3390/children10091574

**Published:** 2023-09-19

**Authors:** Dandan Wu, Chunqi Chang, Jinfeng Yang, Jiutong Luo, Sha Xie, Hui Li

**Affiliations:** 1Faculty of Education and Human Development, The Education University of Hong Kong, Hong Kong SAR, China; ddwu@eduhk.hk; 2School of Biomedical Engineering, Shenzhen University Medical School, Shenzhen University, Shenzhen 518055, China; cqchang@szu.edu.cn (C.C.); yangjinfeng2017@email.szu.edu.cn (J.Y.); 3Faculty of Education, Shenzhen University, Shenzhen 518060, China; jtluo@bnu.edu.com (J.L.); xiesha@szu.edu.cn (S.X.); 4Faculty of Education, Beijing Normal University, Beijing 100875, China

**Keywords:** cognitive shifting, fNIRS evidence, modeling hemodynamic changes, *dimensional change card sort* (*DCCS*) task, preschoolers

## Abstract

General linear modeling (GLM) has been widely employed to estimate the hemodynamic changes observed by functional near infrared spectroscopy (fNIRS) technology, which are found to be nonlinear rather than linear, however. Therefore, GLM might not be appropriate for modeling the hemodynamic changes evoked by cognitive processing in developmental neurocognitive studies. There is an urgent need to identify a better statistical model to fit into the nonlinear fNIRS data. This study addressed this need by developing a quadratic equation model to reanalyze the existing fNIRS data (*N* = 38, *M_age_* = 5.0 years, *SD* = 0.69 years, 17 girls) collected from the *mixed-order design Dimensional Change Card Sort* (DCCS) task and verified the model with a new set of data with the *Habit-DisHabit* design. First, comparing the quadratic and cubic modeling results of the *mixed-order design* data indicated that the proposed quadratic equation was better than GLM and cubic regression to model the oxygenated hemoglobin (HbO) changes in this task. Second, applying this quadratic model with the *Habit-DisHabit design* data verified its suitability and indicated that the new design was more effective in identifying the neural correlates of cognitive shifting than the *mixed-order* design. These findings jointly indicate that *Habit-DisHabit Design* with a quadratic equation might better model the hemodynamic changes in preschoolers during the DCCS task.

## 1. Introduction

Near-infrared spectroscopy (NIRS) technology is a portable and comfortable way to measure the hemodynamic changes in targeted brain areas [1,2,3]. It can generate time-sensitive data that can be analyzed using general linear modeling (GLM) to estimate the changes in oxygenated hemoglobin (HbO) and deoxyhemoglobin (HbR) between the task and baseline. For instance, Li et al. examined the effect of heavy tablet use on preschoolers’ executive function during the *Dimensional Change Card Sort* (DCCS) task using functional NIRS (fNIRS) [4]. They conducted *t*-tests and GLM to compare the hemodynamic changes in the non-user and the heavy user groups. They found a significant between-group difference in activating the prefrontal cortex (Brodmann Area 9, BA 9). The ‘Non-user’ activation pattern was ‘*normal and healthy*’, whereas the ‘Heavy-user’ pattern was ‘*not normal and thus needs further exploration*’ [4]. However, they presented no further statistical evidence to demonstrate how ‘*abnormal*’ the ‘*Heavy-user*’ pattern was, as the GLM results provided no details about the local maximum of *HbO* changes and their estimated time, making exact comparisons of the two ways impossible. In addition, the hemodynamic changes over time in brain areas are a nonlinear rather than linear relationship; thus, the GLM analysis conducted by Li et al. [4] might be inappropriate or even inaccurate. To solve this problem, Li et al. [5] proposed a quadratic function to better model the hemodynamic changes of the DCCS task, demonstrating a nonlinear U-shape paradigm. Still, they did not compare the quadratic results against the GLM and the cubic equation analyses. Thus, they could not conclude whether the quadratic equation would be better than GLM and cubic equation results. To fill this gap, this study aimed to re-analyze the same data with a quadratic and cubic equation to identify a better way to model the hemodynamic responses in the DCCS task. Furthermore, the proposed quadratic equation was also applied to analyze a set of DCCS data with a new design to verify its suitability and appropriateness.

### 1.1. Modeling Hemodynamic Changes with GLM

fNIRS technology allows us to monitor brain activation by measuring hemodynamic changes, such as the concentration of *HbO* and *HbR* in targeted brain areas. The *HbO* and *HbR* data are dynamic and changing over time; thus, advanced statistical analyses are needed to examine this type of time-sensitive data [1,2,3]. However, no systematic and standardized approaches were established in the first decade of this millennium; thus, NIRS scientists had the liberty to choose the statistical methods they believed to be appropriate and adequate. Schroeter et al. [6] initially proposed employing *general linear modeling* (GLM) as the standard statistical approach to analyzing fNIRS data. Accordingly, Pouliot et al. [7] concluded that GLM could be used as a legal analysis to examine fNIRS data for spikes and seizures. In 2014, Tak and Ye [8] systematically reviewed the commonly used statistics such as principal component analysis, independent component analysis, false discovery rate, and inference statistics such as the standard *t*-test, F-test, analysis of variance, and statistical parameter mapping framework. Eventually, they proposed adopting the GLM mixed-effect model with restricted maximum likelihood variance estimation to model hemodynamic changes [8].

Since then, employing GLM to estimate hemodynamic changes has become the standard inference statistic for fNIRS data. GLM empowers scientists to assess the subject, channel, and task-specific evoked hemodynamic responses and to robustly separate the evoked brain activity from systemic physiological interference using independent measures of nuisance regressors [1,2,3]. In addition, GLM can significantly enhance the contrast-to-noise ratio of the brain signal, improve feature separability, and ultimately lead to better classification accuracy. In 2015, for example, Bonomini et al. [9] proposed and confirmed a GLM-based new algorithm to statistically estimate the hemodynamic activations, with a K-means method to cluster channels as activated or not activated. Later, Pinti et al. [10] presented a novel analysis method based on the GLM least-squares fit analysis and verified its accuracy and feasibility in modeling fNIRS data in naturalistic environments. Recently, von Lühmann et al. [11] found that GLM could provide better single-trial estimates of brain activity and a new feature type, such as the weight of the individual and channel-specific hemodynamic response function regressor. However, the hemodynamic changes recorded by fNIRS are nonlinear rather than linear; thus, GLM might not be an appropriate statistical method. This study endeavored to identify a better statistical way by comparing quadratic and cubic modeling results of the same DCCS fNIRS data.

### 1.2. Modeling Hemodynamic Changes in the DCCS Task

The DCCS task asks children to sort a set of two-dimensional (i.e., color and shape) test cards (3.5 × 7.0 cm) according to the two target cards that match the former in one dimension but not the other. Then, the children are asked to sort the test cards according to one extent matching the target card (red/blue; shape: boat/rabbit). And the rule for matching is changed according to the experimenter’s instruction (See Figure 1). Initially, Morriguchi and his colleagues [12] conducted fNIRS studies on the DCCS task. They conducted *t*-tests and correlation analyses to compare *HbO* changes between the task and baseline conditions. Recently, Moriguchi and Lertladaluck [13] and Xie et al. [14] conducted the exact *t*-tests and correlation analyses of the *HbO* changes during the same DCCS task. However, the results were contradictory: Moriguchi and Lertladaluck [13] found no significant relationship between prefrontal activations and English proficiency, whereas Xie et al. [14] found a significant correlation. This inconsistency indicated that either the data analysis or the DCCS task paradigm employed by the two teams might need to be revised in identifying the specific neural correlates responsible for cognitive shifting of the DCCS task.

Therefore, first, Li et al. [15] developed the “*habituation–dishabituation paradigm of DCCS task*” (“*Habit-Dishabit Design*” hereafter) and proposed a more direct and critical indicator—the “*V shape by GLM*” to identify cognitive shifting. As shown in Figure 2, this paradigm has improved the arrangement of testing items to maximize the chances of habituation and dishabituation in the participating children. In the pre-switch period (20′), the children were asked to sort six or more cards using the same rule and thus tended to be habituated. Then, they were asked to use the other rule to sort another set of cards (6 or more) in the post-switch period (20′). The three sessions followed the same sorting rule as the second period of the previous session: Session 1: color (6 cards) → shape (6 cards); Session 2: shape (6 cards) → color (6 cards); and Session 3: color (6 cards) → shape (6 cards). The children tended to be habituated when they anticipated that the second round of sorting cards should follow the same rule. In other words, this paradigm helped to trigger the occurrence of habituation and dishabituation. Second, they [15] proposed a pair of GLMs to estimate HbO changes (∆HbO) for the pre- and post-switch periods, using the same regression formula:*Y_ΔHbO_* = *a _pre-switch_* *_or_* *_post-switch_ X _time_* + *b* + *ε*.(1)

In this GLM equation, *X _time_* refers to the response time and *Y predicts* each channel’s hemodynamic changes (∆HbO). A perfect V-shape could be verified if the *a* pre-switch is negative (−a), whereas the *a* post-switch is positive (+a), and both models are significant. The corresponding channel was identified as the neural correlate of ‘cognitive shifting’ [15]. Using this new paradigm, they found a V-shape in BA 6, BA 8, BA 9, BA 10, BA 40, and BA 44, which should be regarded as the neural correlations of cognitive shifting during the DCCS task [15].

### 1.3. The Context of This Study

Recently, Li et al. [4] adopted the mixed-order design DCCS and GLM by Moriguchi and Lertladaluck [12] and Xie et al. [13] to examine the impact of tablet use on preschoolers’ executive function. Using the V-shape by GLM (Equation (1)), they found that the non-users outperformed the heavy users with a significantly higher correct rate in the DCCS task. And the two groups differed significantly in the activation of BA 9 (ch 16), indicating that the Non-user pattern was ‘*normal and healthy*’ [4]. In contrast, the heavy user pattern was ‘*not normal and needs further exploration*’. However, the hemodynamic changes in each channel should be a kind of nonlinear relationship [1,2,3]. In addition, the V-shape by GLM was analyzed and confirmed using a pair of GLMs (Equation (1)): one for the pre-switch period and the other for the post-switch period. This might not be appropriate for modeling the continuous HbO changes evoked by cognitive processing during the DCCS task. The children’s hemodynamic responses are continuous and indivisible during the two periods. Therefore, Li et al. [5] proposed a quadratic function (Equation (2)) to better model the hemodynamic changes of the DCCS task:(2)$$Y_{\Delta HbO}=aX_{time}^{2}+bX_{time}+c+\varepsilon \left(error\;term\right)$$

In particular, ∆*HbO* refers to the HbO changes between the task and baseline, ε is randomly distributed with a mean of zero, and *X_time_* refers to the experiment time [14]. Thus, if *a* > 0, the curve is a typical U-shape; if *a* = 0, the quadratic function does not exist, indicating a linear relationship that GLM could model; and if *a* < 0, the curve is a reversed U-shape.

However, Li et al. [5] did not compare the quadratic results against GLM and the cubic equation analyses, thus failing to identify the best model. Therefore, in this study, we hypothesized that this U-shaped curve by the quadratic function might be more statistically appropriate for modeling hemodynamic changes in the DCCS task than the V-shape by GLM [14]. In addition, we also hypothesized that the “*Habit-DisHabit Design*” might be more appropriate for identifying the neural correlates of cognitive shifting. Accordingly, this study is dedicated to addressing the following questions:Is a quadratic equation better than GLM and cubic equations to model the hemodynamic changes caused by cognitive shifting in the DCCS task?Is *mixed-order design* better than *Habit-DisHabit design* to identify cognitive shifting in the DCCS task?

## 2. Materials and Methods

### 2.1. Participants

This study first re-analyzed the fNIRS data from Li et al. [4], which recruited 38 children (ages 4 to 6.3 years, *M_age_* = 5.0 years, SD = 0.69 years, 17 girls, 21 boys). Please refer to Li et al. [4] for sample details. Then, we applied the quadratic equation to analyze fNIRS data collected from the same sample with a new experiment design: the *Habit-DisHabit* Design DCCS Task. According to Li et al. [4], their parents consented and completed the survey to help identify the heavy users or non-users of tablets at home. Eight children never used tablets; thus, they were included in the ‘Non-user’ group (two girls and six boys). About 16 (12 girls and 4 boys) children were classified into the ‘Heavy-user’ group” because (1) their daily screen time was more than the mean level (M = 17.98 min, SD = 14.29); (2) their tablet use was neither regulated nor limited; and (3) they carried out multiple activities with tablets. In particular, all the participating children were recruited from one public kindergarten in a middle-class Shenzhen community. Their parents and class teachers reported no problems in the children’s neurological or general physical and mental health status. They conducted a post hoc power analysis on G*Power 3.1, using a two-tailed test, a medium effect size (d = 0.50), and an alpha of 0.05 and found that the results could achieve a power of 0.32 [4].

### 2.2. Experimental Paradigm and Instructions

#### 2.2.1. The Mixed-Order Design DCCS Task

This task included 2 target cards and 24 test cards, each different in shape and color. One pair of target trays was used for the three consecutive test sessions, and each session consisted of a rest (20 s) phase and a mix (25 s) phase. During the rest phase, the children were asked to be still, doing nothing. As shown in Figure 1, *the children were asked to sort the cards according to the instructed rule (color or shape) during the mix phase*. The children were given the rule before each trial. Then, in each block, the rule-changing order was fixed and mixed: shape, shape, color, shape, shape, color, shape, shape (a total of 8 cards per block). This fixed order was applied to all the participants to overcome habituation, resulting in more color-to-shape switches in total [4]. For details about this mixed-order design’s experimental paradigm and instructions, please refer to Li et al. [4].

#### 2.2.2. The *Habit-DisHabit* Design DCCS Task

This task is different from the mixed-order design. As shown in Figure 2, the children were asked to sort eight to twelve cards using the same rule within the pre-switch period (20 s). Then, they were asked to use the other rules to sort another eight to twelve cards within the post-switch period (20 s). The three sessions followed the same sorting rule as the second period of the previous session: Session 1: color (20 s) → shape (20 s); Session 2: shape (20 s) → color (20 s); and Session 3: color (20 s) → shape (20 s). This design was inductive to cognitive habituation and dishabituation by repeating the changing rules in the pre-switch period [15]. For details about this design’s experimental paradigm and instructions, please refer to Li et al. (2021) [15].

### 2.3. System and Acquisition

#### 2.3.1. The fNIRS System

In Li et al.’s studies [4,15] and this study, the same multiple-channel fNIRS system (Oxymon Mk III, Artinis, The Netherlands) and child caps were used to simultaneously measure the concentration changes in *HbO*, *HbR*, and total hemoglobin (*HbT*) in the participants. In particular, both studies employed child caps accompanied by the NIRS instrument, which digitized the optode positions, corresponding to Brodmann areas, as shown in Figure 3. An experienced NIRS technician conducted cap placement, hair manipulation and tossing, and optode installation (based on the 10/20 system). This process usually took 10 min, during which the participant was engaged in storybook reading with an experienced preschool teacher. For details, please refer to Li et al. [4].

#### 2.3.2. Data Acquisition

Two wavelengths in the near-infrared range (i.e., 760 nm and 850 nm) were used to measure the changes in optical density and were then converted into changes in the concentration of HbO and HbR using the modified Beer–Lambert law [4,14]. The 17 channels were located following the international 10/20 system for EEG, as shown in Figure 1 of Li et al. (2021) [4], with a 2.5 cm distance between each paired emitter and detector. In particular, the region of interest (ROI) was located at Brodmann areas (BAs) 6/8/9/10/40/44 [4]. In particular, as shown in Figure 3, channels 1 and 9 were located in BA 6, channels 13, 15, and 17 were located in BA 10, channel 10 was located in BA 8, channels 11, 12, 14, and 16 were located in BA 9, channel 4 was located in BA 40, and channels 2, 3, 5, 6, and 7 and 8 were located in the right inferior frontal gyrus (BA 44).

**Figure 3 children-10-01574-f003:**
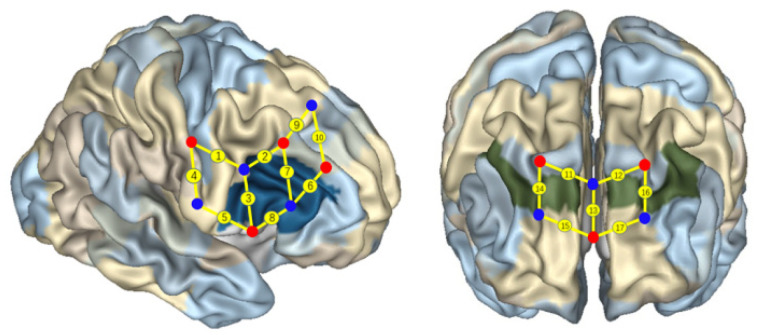
Localization of *regions of interest* [4,6,14]. The red and blue solid circles present the light sources and the probes, respectively. The numbers on the small spheres on the brain map indicate the 17 channels (ch). Channel localization was based on the upper central probe, anchored at Fz according to the international 10–20 system and located at the midpoint between channels 11 (ch 11) and 12 (ch 12). Ch 1 and ch 9 were located in Broadmann area (BA) 6, ch 10 was located in BA 8, ch 11, ch 12, ch 14, and ch 16 were located in BA 9, ch 13, ch 15, and ch 17 were located in BA 10, ch 4 was located in BA 40, and ch 2, ch 3, ch 5, ch 6, ch 7, and ch8 were located in the right IFC (BA 44).

#### 2.3.3. Data Analysis

In Li et al. [4,15] and this study, the mean of the z-scores (*HbO* and *HbR*) was calculated for each DCCS task block separately for each participant. Then, the mean of the z-scores (*HbO* and *HbR*) was calculated by averaging across the three task blocks for each participant. Finally, the means of the z-scores (*HbO* and *HbR*) across all channels were compared using *t*-tests between the ‘Non-user’ and the ‘Heavy-user’ groups using SPSS. Li et al. [4] conducted GLM analysis predicting z-scores (*HbO* and *HbR*) in channel 16 in *R* ($Y_{\Delta HbO}=aX_{time}+b+\varepsilon$) [Equation (1)]. This study explored two sets of polynomial regression to better fit the nonlinear relationship between the hemodynamic changes and the experiment time. The first set was a quadratic equation using *R* [$Y_{\Delta HbO}=aX_{time}^{2}+bX_{time}+c+\varepsilon \left(error\;term\right)$] [Equation (2)]. If *a* > 0, the curve is a typical U-shape; if *a* = 0, the quadratic function does not exist, indicating a GLM linear relationship; and if *a* < 0, the curve is a reversed U-shape. The second set was a cubic equation analysis using the following equation:(3)$$Y_{\Delta HbO}=aX_{time}^{3}+bX_{time}^{2}+cX_{time}+d+\varepsilon \left(error\;term\right)\;\;$$

If *a* = 0, the cubic function does not exist, indicating that a quadratic equation or GLM (if *b* = 0) should be considered. This study would compare the statistical results of the three equations (Equations (1)–(3)) and identify the best-fit model.

## 3. Results

### 3.1. Comparison of the Quadratic and Cubic Modeling Results

First, as shown in Table 1 and Figure 4, the quadratic analysis results for the non-user group indicated that: (1) a U-shaped curve (*a* > 0) was observed in nine channels (ch 1, 2, 5, 6, 7, 8, 9, 11, and 14), and the quadratic model could significantly explain 7.5% to 94.3% of the variance in HbO, *R*^2^s > 0.075, *F*s > 5.94, *p*s < 0.001; (2) the quadratic model did not exist in channels 3 and 4 (a = 0), *R*^2^s > 0.66, *F*s = 145.65, *p*s < 0.001, whereas GLM applied; and (3) a reversed U-shaped curve (a < 0) was observed in channels 10, 12, 13, 15, 16, and 17, *R*^2^s > 0.27, *F*s > 27.27, *p*s < 0.001. The significant *R*^2^s, *F*-values and *p*-values jointly indicated that the U-shaped curve was found for BA 6 (ch 1 and 9), BA 9 (ch 11 and 14), and BA 44 (ch 2, 5, 6, 7, and 8), which were involved in cognitive shifting. Only two channels (ch 3 and 4) could not be estimated by this quadratic equation modeling. Next, a set of cubic equation analyses was conducted for the non-user group, with X^3^ as the cubic term in addition to *X*^2^ as the quadratic term and *X* as the linear term. However, as shown in Table 2, the results for the non-user group indicated that only two channels (ch 11 and 13) could apply to the cubic regression (*a* ≠ 0). All other channels should be modeled with a quadratic equation (*a* = 0). These results indicated that the proposed quadratic equation might be more appropriate than the cubic equation to analyze the DCCS data collected in this study.

Second, as shown in Table 3 and Figure 5, the quadratic analysis results for the heavy user group indicated that: (1) a U-shaped curve (*a* > 0) was observed in channels 2–5, 8, 12, 16, and 17; the quadratic model could significantly explain 37.1% to 92.3% of the *HbO* changes, *R*^2^s > 0.037, *F*s > 43.25, *p*s < 0.001; (2) the quadratic model did not exist in channels 1, 6, 7, and 13 (*a* = 0), *R*^2^s > 0.059, *F*s > 4.68, *p*s < 0.011, whereas the cubic model applied; and (3) a reversed U-shaped curve (*a* < 0) was observed in channels 9–11 and 14–5, *R*^2^s > 0.117, *F*s > 9.87, *p*s < 0.001. The significant *R*^2^s, *F*-values, and *p*-values jointly indicated that the U-shaped curve was found in BA 9 (12 and 16), BA 10 (17), BA 40 (ch 4), and BA 44 (ch 2, 3, 5, and 8). Four channels (ch 1, 6, 7, and 13) could not be estimated by this quadratic equation (see Table 3). In contrast, a set of cubic equation analyses was conducted for the heavy user group, with X^3^ as the cubic term in addition to *X*^2^ as the quadratic term and *X* as the linear term. As shown in Table 4, the results for the heavy user group indicated that no channels could be estimated by the cubic regression as *a* = 0. Therefore, all these results jointly indicated that the proposed quadratic equation might be more appropriate than the cubic equation to analyze the DCCS data collected in this study.

### 3.2. Verification of Quadratic Modeling with New DCCS Data

We applied quadratic modeling with the *Habit-DisHabit* design DCCS data to verify its suitability. As shown in Table 5 and Figure 6, the quadratic regression results for the non-user group indicated that: (1) a U-shaped curve (*a* > 0) was observed in 14 channels (ch 1, 2, 4, 5,7, 8, 9, and 11–17), and the quadratic predictor (experiment time) could explain 48.1% to 96.3% of the variance (HbO), *R*^2^s > 0.48, *F*s > 91.16, *p*s < 0.001; (2) the quadratic function does not exist in channels 3 and 6 (*a* = 0), *R*^2^s > 0.050, *F*s = 5.39, *p*s < 0.005; and (3) a reversed U-shaped curve (*a* < 0) was observed in channels 10, *R*^2^ = 0.570, *F* = 130.73, *p* < 0.001. The significant *R*^2^s, *F*-values, and *p*-values jointly indicated that the U-shaped curve was found for BA 6 (ch 1 and 9), BA 9 (ch 11, 12, 14, and 16), BA 40 (ch 4), and BA 44 (ch 5, 7, and 8), which were involved in cognitive shifting. Only two channels (ch 3 and 6) could not be estimated by this quadratic equation.

Next, as shown in Table 6 and Figure 7, the quadratic modeling results for the heavy user group indicated that: (1) a U-shaped curve (*a* > 0) was observed in channels 1, 2, 4, 8, 9, 11, 12, 14, and 16, and the quadratic predictor (experiment time) could significantly explain 23.3% to 80.3% of *HbO* changes, *R*^2^s > 0.233, *F*s > 29.95, *p*s < 0.001; (2) the quadratic function does not exist in channels 3, 5–7, 10, 13, 15, and 17 (*a* = 0), *R*^2^s > 0.028, *F*s > 2.92, *p*s < 0.05; and (3) no reversed U-shaped curve (*a* < 0) was observed. The significant *R*^2^s, *F*-values, and *p*-values jointly indicated that the U-shaped curve was found in BA 6 (ch 1 and 9), BA 9 (11, 12, 14 and 16), BA 40 (ch 4), and BA 44 (ch 2 and 8). Eight channels (ch 3, 5, 6, 7, 10, 13, 15, and 17) could not be estimated by this quadratic equation.

In summary, the above modeling results for the *mixed-order* and *Habit-DisHabit* design tasks jointly indicated that Equation (2) might be the most suitable model for estimating the hemodynamic changes caused by cognitive shifting. As shown in the summary of Table 7, this model can help identify the most pronounced U-shape in all of the channels observed in the DCCS studies.

## 4. Discussion

### 4.1. Quadratic: More Appropriate Modeling

Previous studies that investigated the HbO changes during the DCCS task [12,13,14] employed *t*-tests and correlation analyses, which are descriptive and correlational methods that depend on the sample size and may produce inconsistent results. For example, Moriguchi and Lertladaluck [13] found no significant effects, while Xie et al. [14] reported a significant correlation. Moreover, these methods could not establish the causal relationship between behavioral and hemodynamic changes in the DCCS task. Thus, Li et al. [15] introduced the “V shape by GLM” (Equation (1)) to model the hemodynamic changes of cognitive shifting, which is an improvement over the previous analytical approaches. However, this linear model does not adequately capture the hemodynamic changes in each channel, as the HbO changes are continuous and wave-like rather than discrete and linear. Therefore, it is not suitable to use GLM to simulate the hemodynamic changes, and a more appropriate statistical model is needed.

Accordingly, this study first reanalyzed the data in Li et al. [4], using both quadratic and cubic equations to model the continuous HbO changes in the DCCS tasks with the mixed-order design. In particular, as shown in Table 2 (the non-user group), only two channels (ch 11 and 13) could be modeled by Equation (3); the other 15 channels were quadratic models. Furthermore, for the heavy user group (Table 4), all channels could not be modeled by Equation (3) (cubic modeling); instead, only quadratic modeling could fit the HbO changes in the 17 channels. As shown in Figure 4 and Figure 5, a comparison of the modeling results indicated that quadratic modeling (Equation (2)) was more effective and appropriate than cubic modeling (Equation (3)).

Next, comparing the quadratic modeling results with the GLM results in Li et al. [4] indicated that nonlinear modeling might be a more sensitive and better fit than linear modeling. In particular, the GLM results in Li et al. [4] demonstrated that BA 9 was significantly activated only in the non-user group during the DCCS task. In contrast, a significant decrease was found for the heavy user group, demonstrating a substantial increase after the twelfth second [4]. Thus, they concluded that BA 9 was significantly activated only in the non-user group during the DCCS task. However, this study reanalyzed the same data using the quadratic equation and found a significant U-shape in this channel (BA 9) for both non-user and heavy user groups, indicating that BA 9 was an essential neural correlate of cognitive shifting. Why could the GLM results not identify the nuance changes in the heavy user group? This is because GLM could only generate a line to demonstrate the general trend. Thus, it could not model the second half of the quadratic curves of the mixed-order DCCS task, especially when there was a U-shape. Therefore, this comparison indicated that the U-shape by a quadratic equation might be more powerful and efficient in identifying the neural correlates of cognitive shifting.

Last, this study also applied Equation (2) with the new *Habit-DisHabit* design data and found a U shape in 14 channels; only one channel (ch 10) had a reversed U shape. This finding indicated that quadratic equations rather than GLM could help identify the neural correlations of cognitive shifting in the DCCS task. Therefore, the quadratic equation (Equation (2)) might be a better model of the hemodynamics of cognitive shifting and should be widely promoted to analyze the DCCS fNIRS data.

### 4.2. Habit-DisHabit Design: More Effective for Identifying Cognitive Shifting

This study first re-analyzed Li et al. [4] data with a quadratic equation and identified the U-shape in 9 channels for the non-user group. Then, an analysis of the new Hab-it-DisHabit design data found a U-shape in 14 channels. Similarly, the re-analysis identified a U-shape in 8 channels within the heavy user group, whereas the new design data demonstrated it in 9 channels. The within-group increases in U shape indicated that the *Habit-DisHabit* design could help identify more correlated channels of cognitive shifting in the DCCS task. In addition, within the non-user group, the mixed-order design data indicated that six channels had a reversed U-shape, indicating that the corresponding channels’ HbO changes increased over time.

In contrast, the *Habit-DisHabit* design data indicated only one channel (ch 10) had a reversed U-shape. Therefore, this accumulative increase cannot reflect the rise and fall of HbO over time corresponding to cognitive shifting. For the heavy user group, the mixed-order design data indicated that five channels had a reversed U-shape, indicating that the HbO changes increased over time. In contrast, the *Habit-DisHabit* design data indicated no channel had a reversed U-shape, indicating a tendency of decreasing HbO in all the channels. This tendency might reflect the unique brain activation pattern of the Heavy user, which will be further explored in future studies.

Most of the fNIRS studies on the DCCS task analyzed the changes in HbO between the task and baseline conditions and, accordingly, could not identify the specific neural correlates responsible for cognitive shifting (CS). There is an urgent need to identify the direct and critical indicator of CS, and the key to this search is ‘habituation’, the fundamental mechanisms underlying human being’s cognition and behavior [15]. When the same stimulus (switching rule) is repeated repeatedly, there will be a reduced response from the exact neural correlates and a decrease in HbO in the blood [16]. Unfortunately, the widely used mixed-order DCCS design [12,13,14] kept changing the switching rules, thus preventing children from habituating their responses. Therefore, this design could not generate the habituation–dehabituation process, which could be an observable marker of the CS. Instead, the *Habit-DisHabit* design prompted habituation and dishabituation in the children’s responses; the ‘U-shape’ with quadratic modeling could exactly demonstrate the occurring moment of cognitive shifting for each channel, which is more powerful in identifying the neural correlates of CS. In summary, this comparison indicated that the *Habit-DisHabit* design might be more effective in identifying the neural correlates of and should be widely promoted in the DCCS tasks and should be widely promoted. Nevertheless, further studies with more samples could help verify and improve this new design.

## 5. Conclusions, Limitations, and Implications

First, this study found that quadratic equations [Equation (2)] might be more appropriate for modeling the *HbO* changes in the DCCS tasks by re-analyzing Li et al. [4] and analyzing the new data. Second, this study proved that the *Habit-DisHabit* design DCCS, in conjunction with the quadratic equation, could effectively identify the neural correlates of cognitive shifting.

However, these results must be interpreted cautiously, as the sample size was tiny. The previous studies by Li et al. [4,15] were stopped by the unexpected COVID-19 lockdown in China in late January 2020. Thus, only 38 complete cases were included in this study. In the future, more samples with more age ranges should be involved to further verify this quadratic modeling method and the *Habit-DisHabit* design.

Nevertheless, the findings have some implications for future study and practical improvement. First, the quadratic equation should be considered a standard nonlinear model to estimate hemodynamic changes in the DCCS tasks. Second, the *Habit-DisHabit* design DCCS could be widely used and further developed to identify the neural correlates of cognitive shifting better. Third, the finding that non-users and heavy users had different brain activation patterns implies that further studies should be conducted to examine the impact of pad use on executive function, and we should consider limiting and regulating children’s digital use in the early years. Recently, Eng et al. [17], Kerr-German and Buss [18], Li et al. [19], and the pioneer Moriguchi and his colleagues [20,21] have conducted fNIRS studies on executive function development in young children using traditional GLMs. Even though these studies have advanced our understanding of the neural correlates of executive function, reanalyzing their data using the quadratic modeling method (Equation (2)) will generate some unexpected results that could go deeper into the underlying neuropsychological mechanisms.

## Figures and Tables

**Figure 1 children-10-01574-f001:**
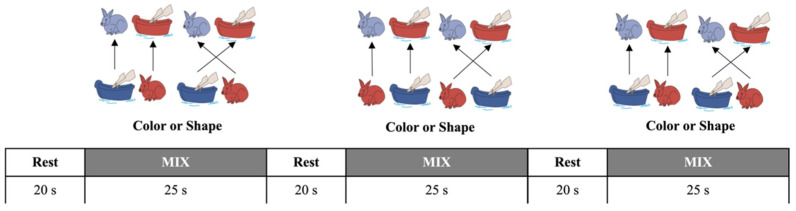
The *mixed-order design DCC*S tasks [4,12,14]. It was a block design with a mixed order of switching rules. The children performed three consecutive test sessions, each consisting of three rest phases (20 s) and three testing phases (25 s). This mixed-order design prevented the children from accurately predicting the switching rules and, thus, could not generate habituation.

**Figure 2 children-10-01574-f002:**
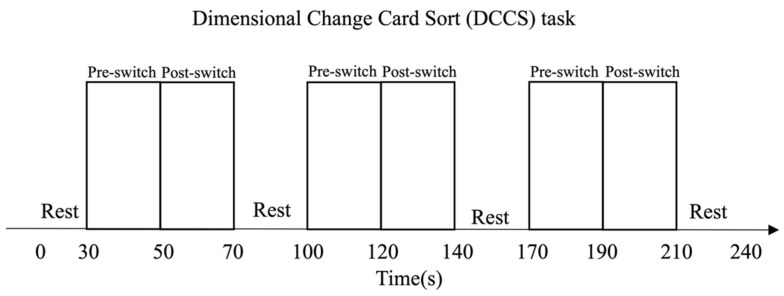
The *Habit-DisHabit Design DCCS* Task [15]. It was a block design with a fixed order of switching rules. The children performed three consecutive test sessions, each consisting of three rest phases (30 s) and three testing phases (40 s). This fixed-order design promoted the children to predict the switching rules accurately and thus habituate their responses.

**Figure 4 children-10-01574-f004:**
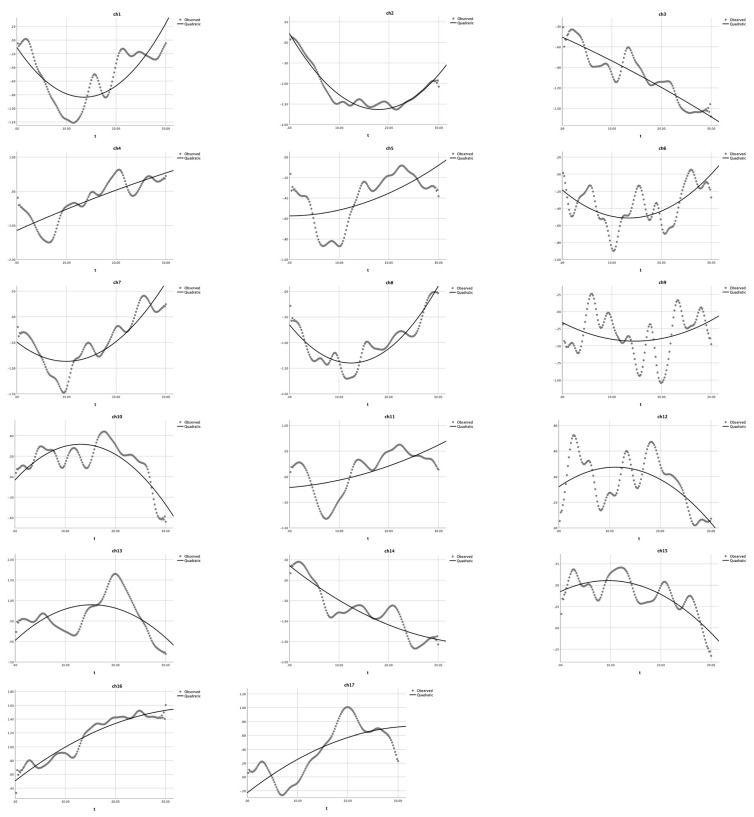
Quadratic modeling the HbO changes in the non-user group. The dotted lines present the observed HbO changes, and the solid lines demonstrate the quadratic curves. The X-axis represents the time (t) and the Y-axis presents the z-scores of HbO changes.

**Figure 5 children-10-01574-f005:**
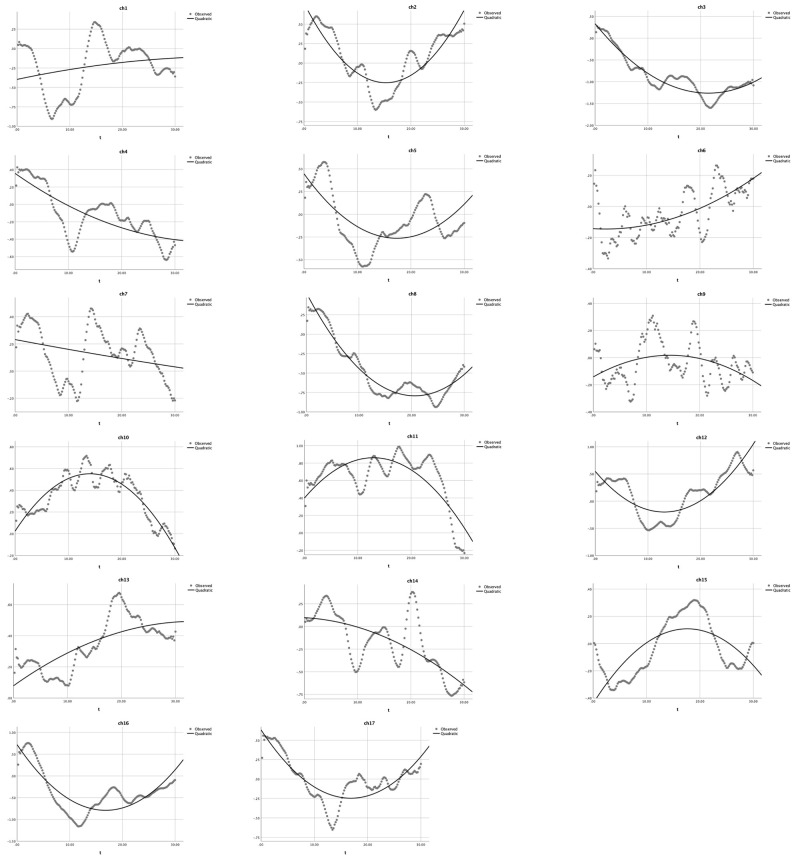
Quadratic equation modeling the HbO changes in the heavy user group during the mixed-order design DCCS tasks [15]. The dotted lines present the observed HbO changes and the solid lines demonstrate the quadratic curves. The X-axis represents the time (t) and the Y-axis presents the z-scores of HbO changes.

**Figure 6 children-10-01574-f006:**
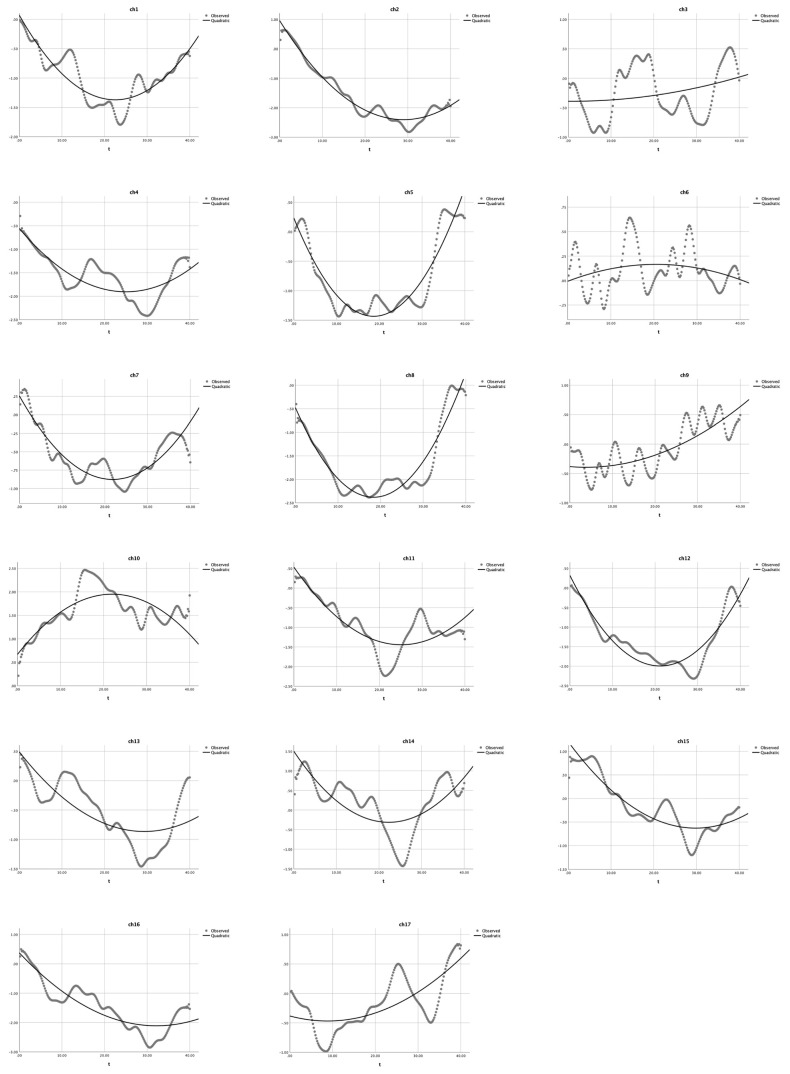
Quadratic equation modeling the HbO changes in the non-user group during the *Habit-DisHabit* DCCS Task. The dotted lines present the observed HbO changes, and the solid lines demonstrate the quadratic curves. The X-axis represents the time (t) and the Y-axis presents the z-scores of HbO changes.

**Figure 7 children-10-01574-f007:**
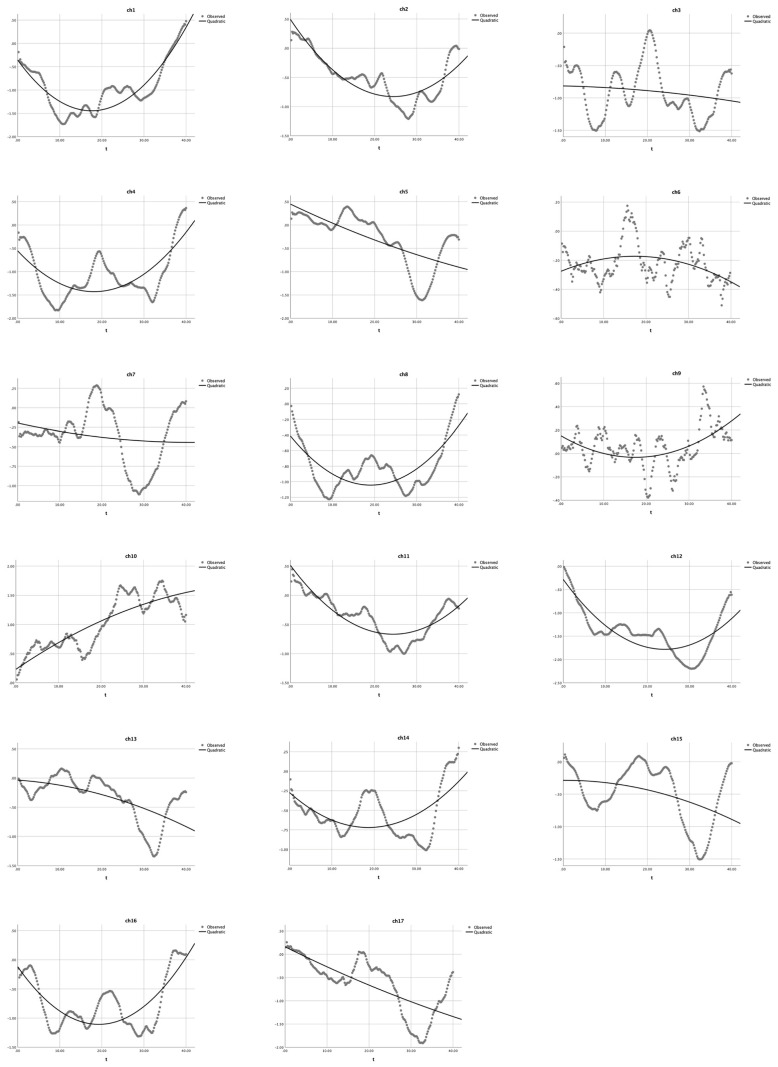
Quadratic equation modeling the HbO changes in the heavy user group during the *Habit-DisHabit* DCCS task. The dotted lines present the observed HbO changes, and the solid lines demonstrate the quadratic curves. The X-axis represents the time (t), and the Y-axis presents the z-scores of HbO changes.

**Table 1 children-10-01574-t001:** The quadratic modeling results for the non-user group.

	Model Summary			Regression Estimates			Quadratic
	*R* ^2^	*F*	Sig.	*c*	*b*	*a*	
Ch 1	0.566	96.041	0	−0.107	−0.108	0.004	√
Ch 2	0.943	1217.46	0	0.219	−0.271	0.011	√
Ch 3	0.846	405.072	0	−0.465	−0.033	0	GLM
Ch 4	0.665	145.654	0	−1.149	0.065	0	GLM
Ch 5	0.358	41.019	0	−0.574	0	0.001	√
Ch 6	0.353	40.048	0	−0.185	−0.049	0.002	√
Ch 7	0.729	198.044	0	−0.492	−0.076	0.004	√
Ch 8	0.832	363.664	0	−0.652	−0.121	0.005	√
Ch 9	0.075	5.941	0.003	−0.158	−0.037	0.001	√
Ch 10	0.636	128.357	0	−0.05	0.08	−0.003	√
Ch 11	0.358	40.957	0	−0.211	0.008	0.001	√
Ch 12	0.352	39.98	0	0.32	0.028	−0.001	√
Ch 13	0.271	27.279	0	0.027	0.114	−0.004	√
Ch 14	0.807	307.87	0	0.357	−0.097	0.001	√
Ch 15	0.624	121.91	0	0.424	0.027	−0.001	√
Ch 16	0.919	830.919	0	0.506	0.057	−0.001	√
Ch 17	0.562	94.382	0	−0.23	0.057	−0.001	√

Note: *Y_HbO change_ = ax*^2^ + *bx* + *c*.

**Table 2 children-10-01574-t002:** The cubic modeling results for the non-user group.

	Model Summary			Regression Estimates				
	*R* ^2^	*F*	Sig.	*d*	*C*	*b*	*a*	Quadratic
Ch 1	0.840	256.361	0.000	0.447	−0.324	0.022	0.000	√
Ch 2	0.985	1096.976	0.000	0.382	−0.271	0.011	0.000	√
Ch 3	0.858	293.571	0.000	−0.384	−0.075	0.003	0.000	√
Ch 4	0.789	181.806	0.000	−0.570	−0.161	0.018	0.000	√
Ch 5	0.835	246.190	0.000	−0.105	−0.183	0.016	0.000	√
Ch 6	0.355	26.737	0.000	−0.157	−0.060	0.003	0.000	√
Ch 7	0.885	372.941	0.000	0.050	−0.288	0.021	0.000	√
Ch 8	0.863	307.565	0.000	−0.439	−0.204	0.012	0.000	√
Ch 9	0.199	12.095	0.000	−0.466	0.083	−0.009	0.000	√
Ch 10	0.789	182.221	0.000	0.266	−0.043	0.007	0.000	√
Ch 11	0.772	165.231	0.000	0.514	−0.276	0.024	−0.001	Cubic
Ch 12	0.440	38.205	0.000	0.481	−0.035	0.004	0.000	√
Ch 13	0.726	129.252	0.000	0.919	−0.235	0.025	−0.001	Cubic
Ch 14	0.858	294.045	0.000	0.720	−0.239	0.013	0.000	√
Ch 15	0.642	87.397	0.000	0.505	−0.004	0.001	0.000	√
Ch 16	0.953	976.936	0.000	0.663	−0.005	0.004	0.000	√
Ch 17	0.890	394.158	0.000	0.356	−0.172	0.018	0.000	√

Note: *Y_HbO change_* = *ax*^3^ + *bx*^2^ + *cx* + *d*.

**Table 3 children-10-01574-t003:** The quadratic modeling results for the heavy user group.

	Model Summary			Regression Estimates			
	*R* ^2^	*F*	Sig.	*c*	*B*	*a*	Quadratic
Ch 1	0.06	4.687	0.011	−0.397	0.014	0	GLM
Ch 2	0.727	196.141	0	0.769	−0.133	0.004	√
Ch 3	0.887	576.348	0	0.332	−0.148	0.003	√
Ch 4	0.595	107.839	0	0.356	−0.043	0.001	√
Ch 5	0.371	43.26	0	0.446	−0.082	0.002	√
Ch 6	0.432	56.012	0	−0.142	−0.002	0	GLM
Ch 7	0.092	7.469	0.001	0.232	−0.007	0	GLM
Ch 8	0.923	880.189	0	0.534	−0.129	0.003	√
Ch 9	0.118	9.88	0	−0.142	0.022	−0.001	√
Ch 10	0.816	325.979	0	0.025	0.075	−0.003	√
Ch 11	0.568	96.717	0	0.399	0.071	−0.003	√
Ch 12	0.669	148.69	0	0.547	−0.114	0.004	√
Ch 13	0.521	79.901	0	0.078	0.025	0	GLM
Ch 14	0.475	66.407	0	0.097	−0.003	−0.001	√
Ch 15	0.477	67.002	0	−0.439	0.062	−0.002	√
Ch 16	0.668	147.668	0	0.719	−0.18	0.005	√
Ch 17	0.683	158.11	0	0.633	−0.104	0.003	√

Note: *Y_HbO change_* = *ax*^2^ + *bx* + *c*.

**Table 4 children-10-01574-t004:** The cubic modeling results for the heavy user group.

	Model Summary			Regression Estimates				
	*R* ^2^	*F*	Sig.	*d*	*c*	*B*	*a*	Quadratic
Ch 1	0.506	49.781	0	0.203	−0.220	0.019	0.000	√
Ch 2	0.732	132.969	0	0.833	−0.158	0.006	0.000	√
Ch 3	0.887	383.652	0	0.362	−0.160	0.004	0.000	√
Ch 4	0.740	138.207	0	0.658	−0.161	0.010	0.000	√
Ch 5	0.603	73.794	0	0.845	−0.238	0.015	0.000	√
Ch 6	0.461	41.568	0	−0.073	−0.029	0.003	0.000	√
Ch 7	0.469	43.028	0	0.555	−0.134	0.010	0.000	√
Ch 8	0.924	593.347	0	0.497	−0.114	0.002	0.000	√
Ch 9	0.123	6.849	0	−0.171	0.033	−0.002	0.000	√
Ch 10	0.829	236.693	0	0.090	0.049	−0.001	0.000	√
Ch 11	0.764	157.399	0	0.743	−0.063	0.008	0.000	√
Ch 12	0.731	132.299	0	0.823	−0.222	0.013	0.000	√
Ch 13	0.821	223.106	0	0.330	−0.073	0.008	0.000	√
Ch 14	0.563	62.819	0	0.361	−0.107	0.008	0.000	√
Ch 15	0.631	83.108	0	−0.219	−0.024	0.005	0.000	√
Ch 16	0.793	186.167	0	1.186	−0.362	0.020	0.000	√
Ch 17	0.739	138.152	0	0.814	−0.175	0.009	0.000	√

Note: *Y_HbO change_* = *ax*^3^ + *bx*^2^ + *cx* + *d*.

**Table 5 children-10-01574-t005:** Quadratic regression predicting *HbO* changes for the non-user group in *the DCCS Habit-DisHabit design* task.

	Model Summary			Parameter Estimates			U-Shape
	*R* ^2^	*F*	Sig.	*c*	*b*	*a*	
Ch 1	0.762	315.838	0.000	0.068	−0.128	0.003	√
Ch 2	0.963	2586.161	0.000	0.960	−0.232	0.004	√
Ch 3	0.076	8.130	0.000	0.390	0.000	0.000	GLM
Ch 4	0.590	141.630	0.000	−0.559	−0.108	0.002	√
Ch 5	0.862	616.488	0.000	0.232	−0.178	0.005	√
Ch 6	0.052	5.391	0.005	−0.007	0.017	0.000	GLM
Ch 7	0.798	388.446	0.000	0.256	−0.104	0.002	√
Ch 8	0.895	841.443	0.000	−0.477	−0.207	0.006	√
Ch 9	0.608	152.895	0.000	−0.383	−0.006	0.001	√
Ch 10	0.570	130.726	0.000	0.669	0.116	−0.003	reversed
Ch 11	0.685	213.853	0.000	0.532	−0.157	0.003	√
Ch 12	0.844	5321.266	0.000	0.318	−0.218	0.005	√
Ch 13	0.541	115.909	0.000	0.489	−0.092	0.002	√
Ch 14	0.497	97.300	0.000	1.501	−0.163	0.004	√
Ch 15	0.827	469.602	0.000	1.201	−0.123	0.002	√
Ch 16	0.784	356.732	0.000	0.374	−0.155	0.002	√
Ch 17	0.481	91.169	0.000	−0.384	−0.019	0.001	√

Note: *Y_HbO change_* = *ax*^2^ + *bx* + *c*.

**Table 6 children-10-01574-t006:** Quadratic regression predicting HbO changes for the heavy user group in the DCCS *Habit-DisHabit* design task.

	Model Summary			Parameter Estimates			U-Shape
	*R* ^2^	*F*	Sig.	*c*	*b*	*a*	
Ch 1	0.803	400.535	0.000	−0.358	−0.124	0.004	√
Ch 2	0.755	304.194	0.000	0.493	−0.108	0.002	√
Ch 3	0.029	2.924	0.056	−0.815	−0.001	0.000	GLM
Ch 4	0.414	69.509	0.000	−0.555	−0.096	0.003	√
Ch 5	0.484	92.488	0.000	0.447	−0.043	0.000	GLM
Ch 6	0.123	13.867	0.000	−0.276	0.012	0.000	GLM
Ch 7	0.040	4.107	0.018	−0.196	−0.012	0.000	GLM
Ch 8	0.473	88.232	0.000	−0.412	−0.066	0.002	√
Ch 9	0.233	29.960	0.000	0.151	−0.021	0.001	√
Ch 10	0.740	280.658	0.000	0.231	0.050	0.000	GLM
Ch 11	0.747	290.877	0.000	0.512	−0.097	0.002	√
Ch 12	0.631	168.348	0.000	−0.284	−0.124	0.003	√
Ch 13	0.381	60.718	0.000	−0.036	−0.005	0.000	GLM
Ch 14	0.274	37.205	0.000	−0.273	−0.048	0.001	√
Ch 15	0.160	18.810	0.000	−0.287	0.000	0.000	GLM
Ch 16	0.523	108.047	0.000	−0.119	−0.103	0.003	√
Ch 17	0.540	115.509	0.000	0.160	−0.045	0.000	GLM

Note: *Y_HbO change_* = *ax*^2^ + *bx* + *c*.

**Table 7 children-10-01574-t007:** Observed U-shape by quadratic modeling in the HbO changes for the non-user and heavy user groups.

DCCS Design	Channel																
	1	2	3	4	5	6	7	8	9	10	11	12	13	14	15	16	17
Non-user in mixed-order	√	√	L	L	√	√	√	√	√	√	√	√	√	√	√	√	√
Non-user in *Habit-DisHabit*	√	√	L	√	√	L	√	√	√	√	√	√	√	√	√	√	√
Heavy user mixed-order	L	√	√	√	√	L	L	√	√	√	√	√	L	√	√	√	√
Heavy user in *Habit-DisHabit*	√	√	L	√	L	L	L	√	√	L	√	√	L	√	L	√	L
Brodmann area (BA)	6	44	44	40	44	44	44	44	6	8	9	9	10	9	10	9	10

Note: √ = U-shape by quadratic; L = linear relationship.

## Data Availability

All the data for this study will be made available upon request.

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
