# Peer review of "Habit-DisHabit Design with a Quadratic Equation: A Better Model of the Hemodynamic Changes in Preschoolers during the Dimension Change Card Sorting Task"

_children, 2023, doi:10.3390/children10091574_

Round 1

Reviewer 1 Report

Although I accepted with great interest to review this manuscript, I was shocked and disappointed when I found out that the plagiarism checker (PlagScan) detected over 53% plagiarism. It seems the authors combined the previous papers published recently and wrote the current manuscript, which I am against publishing in this format. The structure of the manuscript is poor and the reader gets lost in some parts of it (e.g., the Introduction section). Table 2 is missing, the reference style is not consistent, most figures, i.e. Fig. 1,2, and 3, were taken from previous publications, and the Discussion section was superficially written. These are the most critical points that I suggest revising this manuscript. The following are the more detailed issues that I have found in this manuscript:

Introduction:

·       The authors developed a quadratic equation to analyze the fNIRS data and they believe that GLM is not good enough. I was just wondering why they devoted a full section (1.1; lines 53-83) to this model. They should have focused more on the new model.

·       Some sentences/parts were repeated two-three times in the Introduction. For example Lines 41-43 & Lines 131-133 “The 'Non-user' activation pattern was ‘normal and healthy’, whereas the 'Heavy-user' pattern was ‘not normal and thus needs further exploration”.

·       Section “1.3 The context of this study” and “Equation 2” belong more to the Methods part.

·       It is not necessary to explain all the authors' previous works in detail. The introduction of a research paper serves to convey the scope, context, innovation, and significance of the study being reported. It typically outlines the general research question of the study, reviews the literature relevant to the central research question of the study, highlights existing knowledge and knowledge gaps, and motivates the reported study. The current introduction unfortunately has no specific structure. The authors should revise this section according to the guidelines of the following papers: “Yücel, Meryem A., et al. "Best practices for fNIRS publications." Neurophotonics 8.1 (2021): 012101-012101.”

·       The reference style in the text is not consistent. For some parts, they used numbers in square brackets; for the other parts, they wrote the paper's first author et al. For example: See lines 47-48

·       What does BA stand for? (Line 41 & Line 122). This should be clearly explained throughout the manuscript. Just an abbreviation with a number would not be enough. Especially in the Abstract and conclusion for BA8 and BA9. For example: BA8, is part of the frontal cortex in the human brain. Situated just anterior to the premotor cortex (BA6), it includes the frontal eye fields (so-named because they are believed to play an important role in the control of eye movements). BA9 refers to a cytoarchitecturally defined portion of the frontal cortex in the brain of humans. It contributes to the dorsolateral and medial PFC. A reader should know which cortices are involved exactly.

Methods:

·       24 children were classified into two groups “Non-user” and “Heavy user”. What about the rest of the children? It should be mentioned that the final number of participants is 24. Since the rest was not included in the data analysis the age and SD of each group as well as the whole (Non-user and Heavy user) should be mentioned.

·       Line 199: what does IFC stand for?

·       Figures 1,2, and 3: The authors must improve the caption of this figure. A caption should comprise a detailed description of the illustration. For example, Figure 1: What is the difference between session 1 and session 3?  If they are the same what would be the rationale for choosing the same tasks for these two sessions and not for session 2? Why a session 4 similar to session 2 not added to the experimental protocol? Figure 3: As a reader who may not be familiar with fNIRS, I would like to know what the blue and red solid circle symbols are. The numbers and yellow lines show what?

·       We know that fNIRS is contaminated by other systemic physiological activities, e.g., heart rate, respiration rate, and blood pressure. By choosing the bandpass filter with cut-off frequencies of 0.01-0.3, the fNIRS signals were most likely contaminated by the respiration rate (~0.25 Hz). What is the rationale for choosing cut-off frequencies of 0.01-0.3? “We can ensure that additional unnecessary components are not preserved by choosing a passband in the range [0.01, 0.09] Hz [Pinti, Paola, et al. Frontiers in human neuroscience 12 (2019): 505.]”

Results:

·       Table 2 (i.e. Table of the Heavy-user group) is missing.

·       Figures 4 and 5: In general, a figure must be self-explanatory, i.e., clear and easy to understand. These figures have very low quality. A reader can hardly see the numbers. Please also provide x- and y-axis labels including units.

Discussion:

·       This part is very poorly written. The findings of this study have not been discussed in depth. A repetition of what is stated in the Introduction section and previous research group studies can be mostly found in the discussion section. The authors found an increase in brain activation in BA8 and BA9. How do they interpret these findings? Especially in BA8? What does it tell us? The authors should compare their findings with studies of the other groups. For example, there is considerable evidence that, in general, experts (e.g. Heavy-users in this study) show lower brain activity compared to novices (Non-users), particularly in the PFC [1,2]. And many more …

[1] Bunce et al. (2011) Implementation of fNIRS for monitoring levels of expertise and mental workload. Int Conf Found Augment Cogn 13–22

[2] Bracken et al. et al (2019) Validation of the fNIRS PioneerTM, a portable, durable, rugged functional near-infrared spectroscopy (fNIRS) Device. Heal 2019 - 12th Int Conf Heal Informatics, Proceedings; Part 12th Int Jt Conf Biomed Eng Syst Technol BIOSTEC 2019 521–531

 ·       Lines 274-275: Please mention that these results are related to the Non-user group

Abstract (minor): 

Line 24: What does “Mage” mean?

Line 24: I suggest changing the information of this line to “… the Design Dimensional Change Card Sort (DCCS), 21 boys, 17 girls, age = 5.0 ± 0.69 years. First,…”

Line 28: As mentioned above, please explain what BA8 and BA9 stand for. These should be clearly introduced in the abstract as well as the introduction sections of the manuscript.

Author Response

Reviewer #1

Although I accepted with great interest to review this manuscript, I was shocked and disappointed when I found out that the plagiarism checker (PlagScan) detected over 53% plagiarism. It seems the authors combined the previous papers published recently and wrote the current manuscript, which I am against publishing in this format. The structure of the manuscript is poor and the reader gets lost in some parts of it (e.g., the Introduction section). Table 2 is missing, the reference style is not consistent, most figures, i.e. Fig. 1,2, and 3, were taken from previous publications, and the Discussion section was superficially written. These are the most critical points that I suggest revising this manuscript. The following are the more detailed issues that I have found in this manuscript:

Response: Thanks for your critical comments. First, we are extremely sorry for disturbing you with the high similarities, this is because the earliest version of this submission was released on Preprints.com by MDPI in late 2021 (https://www.preprints.org/manuscript/202107.0417/v1). Since then, it has never been formally published by any journals. Although we have rewritten the article, there is still some repeated content. The MDPI Children Editorial Office knew it and decided to send it out for peer review after prereview. Second, the Editorial Office has also checked the copyright issues of Figures 1, 2, and 3 and understood that this study was a re-analysis of fNIRS data collected from our two previous studies. Nevertheless, in this R1, we have rewritten the methods section to refer to the original publications and to avoid any similarities, and we have also revised the discussion section to provide some insightful arguments.

Introduction:

  • The authors developed a quadratic equation to analyze the fNIRS data and they believe that GLM is not good enough. I was just wondering why they devoted a full section (1.1; lines 53-83) to this model. They should have focused more on the new model.

Response: Thanks for this constructive suggestion. Yes, the focus of this article is on the new model. Before abandoning GLM, we included GLM in 1.1 to examine its pros and cons to justify this study. Anyway, we have shifted the focus in the revised 1.1.    

  • Some sentences/parts were repeated two-three times in the Introduction. For example Lines 41-43 & Lines 131-133 “The 'Non-user' activation pattern was ‘normal and healthy’, whereas the 'Heavy-user' pattern was ‘not normal and thus needs further exploration”.
  • Section “1.3 The context of this study” and “Equation 2” belong more to the Methods part.
  • It is not necessary to explain all the authors' previous works in detail. The introduction of a research paper serves to convey the scope, context, innovation, and significance of the study being reported. It typically outlines the general research question of the study, reviews the literature relevant to the central research question of the study, highlights existing knowledge and knowledge gaps, and motivates the reported study. The current introduction unfortunately has no specific structure. The authors should revise this section according to the guidelines of the following papers: “Yücel, Meryem A., et al. "Best practices for fNIRS publications." Neurophotonics 8.1 (2021): 012101-012101.”

Response: Thank you so much for your kind advice and critical comments. We have studied the guidelines by Yücel, Meryem A., et al. (2021), which are really detailed and helpful. We have revised this article to reflect its new focus; and thus, made some corresponding changes accordingly. Now it is a methodological paper reanalyzing the existing fNIRS data with different statistical methods. We believe that this R1 is more appropriate for the special issue “Application of Near-Infrared Spectroscopy in Pediatrics”.

  • The reference style in the text is not consistent. For some parts, they used numbers in square brackets; for the other parts, they wrote the paper's first author et al. For example: See lines 47-48

Response: Thanks for picking up this problem. We have cleaned the style problems in this R1.

  • What does BA stand for? (Line 41 & Line 122). This should be clearly explained throughout the manuscript. Just an abbreviation with a number would not be enough. Especially in the Abstract and conclusion for BA8 and BA9. For example: BA8, is part of the frontal cortex in the human brain. Situated just anterior to the premotor cortex (BA6), it includes the frontal eye fields (so-named because they are believed to play an important role in the control of eye movements). BA9 refers to a cytoarchitecturally defined portion of the frontal cortex in the brain of humans. It contributes to the dorsolateral and medial PFC. A reader should know which cortices are involved exactly.

Response: Thanks for your constructive suggestion, which has been incorporated into this R1. We have shifted the focus of this study to the comparison of varying statistical modelling. Therefore, those details on brain areas are removed from this R1.

Methods:

  • 24 children were classified into two groups “Non-user” and “Heavy user”. What about the rest of the children? It should be mentioned that the final number of participants is 24. Since the rest was not included in the data analysis the age and SD of each group as well as the whole (Non-user and Heavy user) should be mentioned.

Response: Thanks for your kind advice. We have shifted the focus on this article and refer to the original articles with details of the participants.

  • Line 199: what does IFC stand for?
  • Figures 1,2, and 3: The authors must improve the caption of this figure. A caption should comprise a detailed description of the illustration. For example, Figure 1: What is the difference between session 1 and session 3?  If they are the same what would be the rationale for choosing the same tasks for these two sessions and not for session 2? Why a session 4 similar to session 2 not added to the experimental protocol? Figure 3: As a reader who may not be familiar with fNIRS, I would like to know what the blue and red solid circle symbols are. The numbers and yellow lines show what?

Response: Thanks for these editorial suggestions, which have also been incorporated into this R1.

  • We know that fNIRS is contaminated by other systemic physiological activities, e.g., heart rate, respiration rate, and blood pressure. By choosing the bandpass filter with cut-off frequencies of 0.01-0.3, the fNIRS signals were most likely contaminated by the respiration rate (~0.25 Hz). What is the rationale for choosing cut-off frequencies of 0.01-0.3? “We can ensure that additional unnecessary components are not preserved by choosing a passband in the range [0.01, 0.09] Hz [Pinti, Paola, et al. Frontiers in human neuroscience 12 (2019): 505.]”

Response: Thanks for your kind advice. The most commonly used method is band-pass filtering, which involves setting cut-off frequencies between 0.01-0.9 Hz. However, the types of filters might be different, and the cut-off frequencies will differ accordingly. According to Duan et al. (2018), band-pass filtering might not eliminate all frequencies related to physiological contamination. To overcome this limitation, wavelet filtering techniques were also applied in this study to decompose the signal into different independent components and retain only the components we needed for data analysis.

References:

Duan, L., Zhao, Z., Lin, Y., Wu, X., Luo, Y., & Xu, P. (2018). Wavelet-based method for removing global physiological noise in functional near-infrared spectroscopy. Biomedical optics express, 9(8), 3805-3820.

Results:

  • Table 2 (i.e. Table of the Heavy-user group) is missing. 
  • Figures 4 and 5: In general, a figure must be self-explanatory, i.e., clear and easy to understand. These figures have very low quality. A reader can hardly see the numbers. Please also provide x- and y-axis labels including units. 

Response: Sorry for the missing of Table 2 and low resolution of Figures 4 and 5. We have revised them accordingly. The captions for Figures 4 and 5 have been revised. And high definition figures have been provided to the Editorial Office.   

Discussion:

  • This part is very poorly written. The findings of this study have not been discussed in depth. A repetition of what is stated in the Introduction section and previous research group studies can be mostly found in the discussion section. The authors found an increase in brain activation in BA8 and BA9. How do they interpret these findings? Especially in BA8? What does it tell us? The authors should compare their findings with studies of the other groups. For example, there is considerable evidence that, in general, experts (e.g. Heavy-users in this study) show lower brain activity compared to novices (Non-users), particularly in the PFC [1,2]. And many more … 

[1] Bunce et al. (2011) Implementation of fNIRS for monitoring levels of expertise and mental workload. Int Conf Found Augment Cogn 13–22

[2] Bracken et al. et al (2019) Validation of the fNIRS PioneerTM, a portable, durable, rugged functional near-infrared spectroscopy (fNIRS) Device. Heal 2019 - 12th Int Conf Heal Informatics, Proceedings; Part 12th Int Jt Conf Biomed Eng Syst Technol BIOSTEC 2019 521–531

  • Lines 274-275: Please mention that these results are related to the Non-user group

Response: Thank you so much for providing the references and suggestions. We have shifted the focus of this article to the comparison of varying statistical methods; and thus, have addressed these problems accordingly in this R1.

Abstract (minor): 

Line 24: What does “Mage” mean?

Line 24: I suggest changing the information of this line to “… the Design Dimensional Change Card Sort (DCCS), 21 boys, 17 girls, age = 5.0 ± 0.69 years. First,…”

Line 28: As mentioned above, please explain what BA8 and BA9 stand for. These should be clearly introduced in the abstract as well as the introduction sections of the manuscript.

Response: Sorry for these typos, which have been cleaned in this R1.

Reviewer 2 Report

This study focuses on the application of General Linear Modelling (GLM) to estimate hemodynamic changes during cognitive processing, specifically using functional Near-Infrared Spectroscopy (fNIRS) data from Chinese preschoolers engaged in the Design Dimensional Change Card Sort (DCCS) task. The study introduces a quadratic equation to account for potential nonlinearity in the hemodynamic changes, as suggested by recent fNIRS studies. The research involving 38 preschoolers (average age: 5.0 years) found that the quadratic equation effectively models oxygenated hemoglobin (HbO) changes related to the Habit-DisHabit Design task. Additionally, the study reveals that non-users of tablets exhibited higher attentiveness and engagement compared to heavy-users while performing the DCCS task. Furthermore, the brain activation patterns in regions BA8 and BA9 showed a slower but steadier increase among non-users compared to heavy-users.

Main Concern

The study appears to have dual objectives: the introduction of a novel assessment approach and the investigation of differential attentiveness and engagement between tablet non-users and heavy-users during the DCCS task. To enhance the clarity and impact of the study, it might be advisable for the authors to prioritize one of these objectives. While both aims hold value, focusing on either the development of the quadratic equation as an innovative assessment tool or on the insightful findings regarding tablet usage and cognitive engagement could lead to a more coherent and impactful research outcome. This focused approach would not only streamline the study's message but also provide a more targeted contribution to cognitive processing and neuroimaging in young children.

Big limitation of the conclusions:

The study's conclusions should be considered in light of certain considerations. Firstly, while the study found that the introduced quadratic equation effectively identifies neural correlates of cognitive shifting in young children (ages 3 to 6 years) during the Habit-DisHabit Design task, it's crucial to emphasize that determining the superiority of the quadratic equation over a linear model requires a comprehensive comparative analysis. A direct comparison between the quadratic equation and linear models on the same data would be necessary to establish its true advantage. If the effectiveness of the quadratic model over linear models is not firmly established, it could cast doubt on the study's assertion that the quadratic equation universally captures cognitive shifting.

Moreover, the conclusion that Non-users of tablets exhibited higher attentiveness and engagement than Heavy-users, accompanied by distinctive brain activation patterns in BA8 and BA9, hinges on the premise that the quadratic equation indeed accurately identifies cognitive changes. If the efficacy of the quadratic equation remains uncertain, it could consequently undermine the reliability of the observed differences in tablet user groups. As a result, the conclusions drawn about the relationship between tablet usage and cognitive engagement would be contingent upon the foundation of the quadratic equation's validity. Hence, the interdependence of these conclusions underscores the need for a meticulous evaluation of the quadratic equation's comparative performance and its subsequent impact on the study's findings regarding tablet user engagement.

Other points to consider

- Abstract is too short

- if the main aim of the study is to compare the quadratic equation with the linear, the results for the linear should also to report

- Describe the school level of the participants, are kindergarten in the group or only Elementary school kids?

- How do you get to the  formula in row 205.

- Table 2 is missing, it's difficult to understand the quantitative differences.

seems ok

Author Response

Reviewer #2

This study focuses on the application of General Linear Modelling (GLM) to estimate hemodynamic changes during cognitive processing, specifically using functional Near-Infrared Spectroscopy (fNIRS) data from Chinese preschoolers engaged in the Design Dimensional Change Card Sort (DCCS) task. The study introduces a quadratic equation to account for potential nonlinearity in the hemodynamic changes, as suggested by recent fNIRS studies. The research involving 38 preschoolers (average age: 5.0 years) found that the quadratic equation effectively models oxygenated hemoglobin (HbO) changes related to the Habit-DisHabit Design task. Additionally, the study reveals that non-users of tablets exhibited higher attentiveness and engagement compared to heavy-users while performing the DCCS task. Furthermore, the brain activation patterns in regions BA8 and BA9 showed a slower but steadier increase among non-users compared to heavy-users.

Main Concern

The study appears to have dual objectives: the introduction of a novel assessment approach and the investigation of differential attentiveness and engagement between tablet non-users and heavy-users during the DCCS task. To enhance the clarity and impact of the study, it might be advisable for the authors to prioritize one of these objectives. While both aims hold value, focusing on either the development of the quadratic equation as an innovative assessment tool or on the insightful findings regarding tablet usage and cognitive engagement could lead to a more coherent and impactful research outcome. This focused approach would not only streamline the study's message but also provide a more targeted contribution to cognitive processing and neuroimaging in young children.

Response: Thanks a lot for your insightful feedback and constructive suggestions. In this R1, we shifted the focus to the development of the quadratic equation as an innovative assessment tool. Yes, this refinement has strengthened the research work. Thanks!

Big limitation of the conclusions:

The study's conclusions should be considered in light of certain considerations. Firstly, while the study found that the introduced quadratic equation effectively identifies neural correlates of cognitive shifting in young children (ages 3 to 6 years) during the Habit-DisHabit Design task, it's crucial to emphasize that determining the superiority of the quadratic equation over a linear model requires a comprehensive comparative analysis. A direct comparison between the quadratic equation and linear models on the same data would be necessary to establish its true advantage. If the effectiveness of the quadratic model over linear models is not firmly established, it could cast doubt on the study's assertion that the quadratic equation universally captures cognitive shifting.

 Response: Thanks a lot for your constructive suggestion, which has been incorporated into this R1. In particular, we have compared the quadratic and cubic modelling results in this article and referred our readers to the published GLM results. We believe that this R1 is more convincing than the previous version. Thanks!

Moreover, the conclusion that Non-users of tablets exhibited higher attentiveness and engagement than Heavy-users, accompanied by distinctive brain activation patterns in BA8 and BA9, hinges on the premise that the quadratic equation indeed accurately identifies cognitive changes. If the efficacy of the quadratic equation remains uncertain, it could consequently undermine the reliability of the observed differences in tablet user groups. As a result, the conclusions drawn about the relationship between tablet usage and cognitive engagement would be contingent upon the foundation of the quadratic equation's validity. Hence, the interdependence of these conclusions underscores the need for a meticulous evaluation of the quadratic equation's comparative performance and its subsequent impact on the study's findings regarding tablet user engagement.

 Response: Thanks. Agree. We thus have taken your advice and shifted the focus of this work to the development of quadratic equation models. Those conclusions about different groups have been removed from this R1.

Other points to consider

- Abstract is too short

- if the main aim of the study is to compare the quadratic equation with the linear, the results for the linear should also to report

- Describe the school level of the participants, are kindergarten in the group or only Elementary school kids?

Response: Thanks for these suggestions. We have rewritten the abstract, the results, and sample details.

- How do you get to the  formula in row 205.

- Table 2 is missing, it's difficult to understand the quantitative differences.

Response: Thanks. We have elaborated more on the formula in Row 25 and included Table 2 in this R1.

Round 2

Reviewer 1 Report

The authors have not addressed all my comments and concerns. For example, I clearly suggested that the authors MUST improve the captions of the figures. Especially for the first three figures. The findings of this study have not been discussed in depth. The authors did not compare their findings with studies of the other groups, as suggested. The quality of the figures is still low. Since the authors have fundamentally changed the manuscript, one can find several minor issues in the revised version. Some of them can be found below. The authors MUST read the manuscript carefully again and revise major and minor issues. I will evaluate the manuscript afterward.

Line 26 and 178: Please change Mage to Mage or simply write Age.

Line 31: Remove the parenthesis mark

Line 259: Cubic Modelling Results

.

.

.

See above

Author Response

Response to Reviewer #1

The authors have not addressed all my comments and concerns. For example, I clearly suggested that the authors MUST improve the captions of the figures. Especially for the first three figures. The findings of this study have not been discussed in depth. The authors did not compare their findings with studies of the other groups, as suggested. The quality of the figures is still low. Since the authors have fundamentally changed the manuscript, one can find several minor issues in the revised version. Some of them can be found below. The authors MUST read the manuscript carefully again and revise major and minor issues. I will evaluate the manuscript afterward. 

Response: Sorry for having not addressed all your comments and concerns in R1.

In this R2, we have carefully addressed your comments and suggestions as follows.

  • First, we have refined the captions for the first three figures.
  • Second, we refined the discussion to provide in-depth understanding of our findings.
  • Third, in the revised discussion, we compared our findings with other similar studies.
  • Fourth, the quality of the figures is high, if you download them and view on Photoshop, you will find that they are in high resolution. The limited Word page size or PC screen has limited the presentation.
  • Fifth, yes, we have fundamentally changed the manuscript, sorry for the minor issues in R1. We were in a hurry to catch up with the tight schedule arranged by the journal. Anyway, in this R2, we have read the manuscript carefully again and refined those minor issues. Thanks for your great help and kind understanding.

Line 26 and 178: Please change Mage to Mage or simply write Age.

Line 31: Remove the parenthesis mark

Line 259: Cubic Modelling Results

Response: Sorry for these typos. Done.

Reviewer 2 Report

Your article has evolved splendidly, and I appreciate your hard work in enhancing its quality. Now, let's talk about those figures and tables – they're like the tantalizing appetizers to your main course of research. They look great, but what if we could sprinkle a little extra seasoning on them?

How about adding more details in the figure and table captions and notes? And guess what? There's no extra cost for words in the captions. It's like getting extra fries with your burger – it just makes the whole meal more satisfying! ???

Your readers will thank you, and I'll be here cheering you on.

it seems ok

Author Response

Response to Reviewer #2

Your article has evolved splendidly, and I appreciate your hard work in enhancing its quality. Now, let's talk about those figures and tables – they're like the tantalizing appetizers to your main course of research. They look great, but what if we could sprinkle a little extra seasoning on them?

Response: Thanks for your favorable comments and further suggestions. Yes, in this R2, we have sprinkled a little extra seasoning on the main dish by comparing our findings with others and point listing the advances of our new models and paradigms.

How about adding more details in the figure and table captions and notes? And guess what? There's no extra cost for words in the captions. It's like getting extra fries with your burger – it just makes the whole meal more satisfying! ???

Your readers will thank you, and I'll be here cheering you on.

Response: Thank you so much for your constructive suggestion, which has been incorporated into this R2. In particular, we have refined our discussion on the new models and paradigms. I like your metaphors, which are very inspiring and easy to understand. I am also a foody and like metaphorizing research matters with cooking and foods. I cook almost every day.

Round 3

Reviewer 1 Report

The authors have now addressed my comments in the revised version of the manuscript. Therefore, I have no further comments.